# Infarct-like versus Non-Infarct-like Clinical Presentation of Acute Myocarditis: Comparison of Cardiac Magnetic Resonance (CMR) Findings

**DOI:** 10.3390/diagnostics13152498

**Published:** 2023-07-27

**Authors:** Raffaella Capasso, Maria Chiara Imperato, Nicola Serra, Reimy Rodriguez, Maria Rivellini, Massimo De Filippo, Antonio Pinto

**Affiliations:** 1Department of Radiology, CTO Hospital, Azienda Ospedaliera dei Colli, Viale Colli Aminei 21, 80141 Naples, Italymaria.rivellini@ospedalideicolli.it (M.R.); antonio.pinto@ospedalideicolli.it (A.P.); 2Department of Radiology, Santa Maria Incoronata dell’Olmo Hospital, 84013 Cava de’Tirreni, Italy; mcimperato@gmail.com; 3Department of Public Health, University Federico II of Naples, 80131 Naples, Italy; nicola.serra5@gmail.com; 4Department of Medicine and Surgery (DiMec), Section of Radiology, University of Parma, Maggiore Hospital, Via Gramsci 14, 43126 Parma, Italy; massimo.defilippo@unipr.it

**Keywords:** acute myocarditis, infarct-like, Lake-Louise criteria

## Abstract

Background: The clinical presentation of acute myocarditis (AM) is widely variable, ranging from a subclinical disease to an infarct-like syndrome. Cardiac magnetic resonance (CMR) has become the reference non-invasive diagnostic tool for suspected AM, allowing the identification of the various hallmarks of myocardial inflammation (edema, fibrosis, and hyperemia). The aim of the study was to investigate any differences in morphological, functional, and tissue characterization CMR parameters between infarct-like AM in patients with unobstructed coronary arteries and non-infarct-like AM, diagnosed according to the original Lake-Louise Criteria (LLC); Methods: We retrospectively selected 39 patients diagnosed with AM according to LLC, divided into 2 groups according to the clinical presentation: infarct-like in group 1 patients and non-infarct-like in group 2 patients. CMR morphologic, functional, and tissue characterization parameters were analyzed and compared. Results: In group 1, CMR tissue characterization parameters were mainly altereda in septal location with mesocardial myocardial involvement; in group 2, CMR tissue characterization parameters were mainly altered in a lateral location with subepicardial myocardial involvement mainly at the mid-cavity. No significant differences in morphological or functional parameters were observed between the two study groups. Conclusions: Our results showed differences in the localization and distribution of myocardial tissue damage assessed by CMR among forms of AM accompanied by an infarct-like clinical presentation compared with non-infarct-like presentations of AM. The mechanisms underlying the different preferential sites of damage observed in our study are not known, and we do not exclude the possibility of their prognostic implications.

## 1. Introduction

The clinical presentation of acute myocarditis (AM) is widely variable, ranging from a subclinical disease characterized by flu-like symptoms and atypical chest pain to an infarct-like syndrome with severe and/or recurring chest pain, electrocardiogram (ECG) pathologic changes, and elevated troponin (Tn) levels mimicking an acute myocardial infarction, or to fulminant heart failure, cardiogenic shock, and sudden death related to the new onset of arrhythmias and complete heart block [1,2,3]. In a small (7–15%) subgroup of patients presenting with an infarct-like syndrome, subsequent coronary angiography reveals normal or non-flow-limiting coronary artery disease [1,2,3,4,5,6,7,8,9]. These patients presenting with suspected acute coronary syndrome (ACS) and unobstructed coronary arteries pose a difficult clinical and diagnostic dilemma; therefore, cardiac magnetic resonance (CMR) may be particularly useful in determining the diagnosis [1,2,3,4,5,6,7,8,9,10,11]. CMR diagnostic criteria—original Lake Louise criteria (LLC)—for AM have been extensively validated in the literature and are commonly applied in clinical routines with high diagnostic accuracy and sensitivity [1,10,11,12]. In this clinical setting, CMR has become the reference non-invasive diagnostic tool for suspected AM, allowing the identification of the various hallmarks of myocardial inflammation (edema, fibrosis, and hyperemia) and combining its peculiar tissue characterization capabilities with the assessment of biventricular regional and global function [1,11,12]. The aim of the study was to investigate any differences in morphological, functional, and tissue characterization-CMR parameters between infarct-like AM in patients with unobstructed coronary arteries and non-infarct-like AM, diagnosed according to the original LLC.

## 2. Materials and Methods

### 2.1. Patients

The study is a retrospective analysis of patients with a clinical diagnosis of AM confirmed by CMR, according to the original LLC. By a keyword search on the radiology information system (RIS) of our institution, we retrospectively identified CMR reports of patients admitted to our institution during a 3-year period—before the COVID-19 pandemic outbreak—that matched the term “myocarditis”. For each identified patient, the diagnosis at discharge was used to select patients diagnosed with AM. Then clinical, ECG, and laboratory data of AM patients were recorded, and the corresponding CMR images were reviewed in order to select the adequately performed examinations that allowed the CMR diagnosis of AM according to the original LLC. Were excluded from CMR studies that did not meet at least two of the three LLCs. According to clinical presentation, patients were divided into 2 groups to separate patients with the infarct-like syndrome (group 1), characterized by elevated troponin levels, chest pain, and pathological ECG changes, from those with non-infarct-like AM (group 2). All patients included in group 1 had undergone emergency coronary angiography at admission.

### 2.2. CMR Imaging

All studies were performed with a 1.5-T MRI scanner (Philips Achieva; Philips Medical Systems; Best, The Netherlands) using a 16-channel phased-array coil with standard ECG triggering. Cine steady-state free precession (cine- SSFP) CMR images were acquired during breath-holds in the short-axis (SA), 2-chamber, and 4-chamber planes; on short-axis images, the left ventricle was completely encompassed from the atrioventricular ring to the apex, acquiring a total of 10 to 12 images. Morphologic evaluation was performed with T2-weighted (T2w) images and short tau inversion recovery T2w (T2w-STIR) images in long- and short-axis planes, using a triple inversion recovery preparation module, in order to suppress fatty tissue signal and emphasize tissue and myocardial edema/inflammation. A single-shot, spoiled, gradient-echo sequence with saturation prepulse (dynamics turbo field gradient-echo) was used for first-pass perfusion imaging with these typical settings: five non-contiguous slices in short-axis view placed to cover the left ventricular basal to apical planes and recorded continuously for each cardiac cycle; total duration, 1 min with partial breath-hold during initial myocardial enhancement. The acquisition was synchronized to the intravenous injection of 0.2 mmol/kg of gadolinium chelate (gadobenate dimeglumine, Gd-DTPA, Multihance, Bracco, Milan, Italy) at a rate of 3 mL/s, followed by a flush infusion at the same rate. Early post-contrast cine-SSFP for assessing both volume and function of the left ventricles and early myocardial enhancement were acquired in short-axis slices, from the mitral valve plane to the apex. Finally, late gadolinium enhancement (LGE) images were acquired 10 min after contrast medium administration in short-axis, 2-chamber, and 4-chamber planes using an inversion-recovery gradient echo sequence T1-weighted (T1w). Inversion times were adjusted to null normal myocardium (typically 250–300 ms; pixel size 1.7 × 1.4 mm). LGE images were phase-swapped to exclude artifacts.

### 2.3. CMR Analysis

Two experienced radiologists, blinded to study group identity, retrospectively assessed CMR images. Any discrepancies were resolved through discussion until a consensus was reached. For image analysis, the left ventricular (LV) myocardium was divided into 17 segments according to the American Heart Association (AHA) classification [13,14]. LV volumes and function were measured using standard techniques and specific software for cardiac analysis (ViewForumR6.3, Philips Medical System, Erlangen, Germany, or ARGUS Flow, Siemens Healthineers, Milano, Italy). For the evaluation of LV global and regional function and the calculation of LV mass, the endocardial and epicardial borders were manually drawn in the end-diastolic and end-systolic short-axis cine-SSFP images; diastole and systole were defined, respectively, as the points of maximum and minimum size of the LV, determined by the average ventricular short axis. Papillary muscles and trabeculations were not included in the myocardium. LV end-diastolic volume (EDV), LV end-systolic volume (ESV), ejection fraction (EF), and LV mass were determined. Functional LV parameters were normalized to the body surface area of each patient.

The presence, pattern, size, extension, and distribution of myocardial parietal edema, early gadolinium enhancement (EGE) expression of hyperemia, and LGE were assessed. Myocardial edema on T2-weighted STIR images was calculated by using the T2-ratio method of quantification by manually outlining two separate regions of interest, respectively, within the entire LV myocardium and the visible skeletal muscle (serratus anterior, combination of teres minor and infraspinatus, subscapularis, extensor of the spine, longissimus dorsi, or a combination of major and minor pectoralis depending on structure visibility and signal intensity homogeneity). Areas with T2 ratio values ≥2 standard deviations (SD) were considered edematous [1,13]. The EGE, defined as an increased normalized gadolinium-DTPA accumulation in the myocardium during the early washout period (about 3–4 min), was evaluated on early post-contrast cine-SSFP images according to the Perfetti et al. method [15]. Perfetti et al. introduced this alternative method for the assessment of hyperemia, relying on relative signal hyperintensity in diastole in SSFP images acquired immediately after contrast administration, as generally occurs in current practice [12]. Hyperemia was identified as the presence of areas of myocardial hyperintensity in SSFP images during the cardiac cycle. Logarithmic colorimetric maps were used to facilitate EGE assessment. The LGE, defined as an area with hyperintense signal intensity >3 times the standard deviation (SD) compared with the intensity of the myocardium of reference, was researched using the IR T1-weighted sequences [1,13]. It was classified as linear or patchy.

In both study groups, for each parameter (edema, hyperemia, and LGE), the number (and percentage) of affected patients and the number (and percentage) of AHA segments involved were calculated. For each parameter (edema, hyperemia, and LGE), the pattern of myocardial involvement is distinguished into subepicardial, intramural, subepicardial with intramural extension, and subendocardial (Figure 1). The number and percentage of affected segments relative to the total affected segments were calculated. To assess the extent and distribution of myocardial damage, the LV was divided into 3 SA planes along the long-axis of the LV: basal, mid-cavity, and apical, using the papillary muscles as anatomic landmarks to distinguish the mid-cavity SA slices from the apical and basal slices. These SA planes were then divided radially into six segments for the basal (1–6 segments) and mid-cavity (7–12 segments) slices, and four segments for the apical slice (13–16 segments), while the 17th segment is the apex itself. AHA segments were then grouped into anterior (segments n. 1, 7, 13), septal (segments n. 2, 3, 8, 9, 14), inferior (segments n. 4, 10, 15), lateral (segments n. 5, 6, 11, 12, 16), and apex (segment n. 17) [14,16,17].

For each study group, the number of affected segments by edema, hyperemia, and LGE was respectively related to the total number of segments for each site (obtained by multiplying the number of patients by the number of segments in each location as anterior, septal, lateral, inferior, or each plane as basal 1–6 segments, mid-cavity, and apical) to assess the damage extension, and it was also related to the total number of affected segments to evaluate the damage distribution. The extent of the pericardial effusion was also estimated for each patient, considering the number of segments of the LV adjacent to the fluid level and expressing it in percentage form of the total segments of the LV.

### 2.4. Statistical Analysis

Data were analyzed using Matlab statistical toolbox version 2008 (MathWorks, Natick, MA, USA) for 32-bit Windows for both study groups: group 1 (infarct-like AM) and group 2 (non-infarct-like AM). Qualitative variables are expressed as numbers and percentages, while quantitative variables are reported as the mean value ± standard deviation (SD). Morphologic, functional, and tissue characterization CMR data were analyzed; normally distributed data were analyzed with the T-Student test, and proportions in 2 groups were compared by the Fisher’s exact test or the Pearson’s chi square test with Yates correction test according to sample size. A *p* value lower than 0.05 was considered statistically significant.

## 3. Results

### 3.1. Patients

We identified 42 patients discharged with an AM diagnosis. Evaluating the technical quality of CMR examinations, we finally selected 39/42 patients diagnosed with AM, according to LLC. Baseline characteristics of selected patients are summarized in Table 1. Group 1 included 14 patients (eight males and six females; mean age 48.15 ± 19.13 years) who presented fever (n.2), chest pain (n.14), dyspnea (n.5), palpitations (n.3), ECG changes including ST segment elevation (n.9) or depression (n.5), T-wave inversion (n.1), and bundle branch block (n.1), associated with serum markers for myocardial necrosis such as creatine kinase-MB (CK-MB) in 13/14 patients and troponin I (Tn-I) in 14/14 patients.

At presentation, 3/14 patients had a history of recent (<6 weeks) infection (1 pharyngitis, 1 colitis, 1 influenza syndrome). The serological tests for the detection of cardiotropic viruses (Adenovirus, Coxsackie B, CMV, EBV, HSV 1–2, Parvovirus B19) revealed the presence of IgM for Coxsackie B in one patient and for HSV1 in one patient. Throat swabs were negative for common pathogens.

Selective angiograms of the left and right coronary arteries were acquired to exclude CAD. The mean time to a CMR scan after invasive coronary angiography was 2.89 ± 3.19 days. Group 2 included 25 patients (17 males and eight females; mean age 36.92 ± 14.14 years) who presented fever (n.9), chest pain (n.16), dyspnea (n.16), and palpitations (n.5) with ECG changes (n.17) and/or serum markers for myocardial damage (n.23) not suspected for acute coronary syndrome.

From the anamnesis, 8/25 patients presented a recent (<2 weeks) history of flu syndrome, 5/25 patients reported a recent (<6 weeks) upper respiratory tract infection and 3/25 patients reported a recent gastroenteritis (<6 weeks). The serological tests for the detection of cardiotropic viruses (Adenovirus, Coxsackie B, CMV, EBV, HSV 1–2, Parvovirus B19) were negative. Throat swabs were positive for the influenza virus in two patients.

The mean time for a CMR scan after admission for acute symptoms was 5.28 ± 6.23 days. The mean age of patients in Group 1 was statistically higher (*p* = 0.0014) than in Group 2.

### 3.2. CMR Morphologic and Functional Findings

CMR morphologic and functional findings and their statistical analysis are reported in Table 2.

Most patients (92.86% in group 1 and 84% in group 2) presented normal LV diameter and thickness without significant differences in these parameters between the two study groups. In group 1, one patient showed LV dilatation (59 mm) and one patient presented focal thickening (14 mm) of the LV myocardium; in group 2, 4 patients showed LV dilatation (60.5 ± 2.82 mm) and two patients presented focal thickening (15 ± 0 mm) of the LV myocardium.

No significant differences in LV volumes, EF, or LV myocardial mass were found between groups 1 and 2. LVEDV was elevated in 2 patients in group 1 and in 1 patient in group 2. In group 1, most patients (71.43%) showed normal (>55%) FE, 2 patients had EF (<45% < EF < 55%) reduction, and two patients had more severe EF (<45%) reduction. In group 2, most patients (72%) showed normal (>55%) EF, 4 patients had EF (<45% < EF < 55%) reduction, and three patients had more severe EF (<45%) reduction; EF was inversely related (*p* = 0.0020) to LGE. LV wall motion was normal in most (71.43% group 1; 80% group 2), while regional wall motion abnormalities were seen in two patients (49%) of group 1, and global hypo-contractility was observed in two patients of group 1 and in two patients of group 2, without significant differences in these parameters between the two study groups. Pericardial effusion was found in 42.9% and 40% of patients in groups 1 and 2, respectively, without a significant difference, although in group 1 it affected more segments (*p* = 0.0365). In both groups, pericardial effusion was mainly located at the lateral LV segments (52.5% in group 1 and 55.1% in group 2).

### 3.3. CMR Tissue Characterization Findings

CMR tissue characterization findings and their statistical analysis are reported in Table 3 and Table 4, respectively.

In group 1, LV myocardial edema was found in all patients, while hyperemia and LGE were not observed in two and one patients, respectively. In group 2, LV myocardial edema, hyperemia, and LGE were not observed in 1, 5, or 1, respectively.

No significant difference in the number of patients with each LV myocardial alteration (edema, hyperemia, and LGE) was found in group 1 compared to group 2. LGE involvement of myocardial segments was significantly lower (*p* = 0.0041) in group 1 than in group 2. No endocardial location of LV myocardial damage was observed in both groups.

Considering subepicardial and intramural location together as a distinctive category of LV myocardial damage, subepicardial-intramural edema was significant (*p* = 0.0467) less frequent in group 1 than group 2; intramural hyperemia was significant (*p* = 1.05 × 10^−3^) more frequent in group 1 than group 2;subepicardial-intramural hyperemia was significant (*p* = 0.0359) less frequent in group 1 than group 2; subepicardial LGE was significant less frequent (*p* = 4.87 × 10^−5^) in group 1 than group 2 while intramural and subepicardial-intramural LGE were significant (respectively *p* = 3.28 × 10^−4^ and *p* = 0.0366) more frequent in group 1 than group 2.

Considering subepicardial and intramural involvement as separate categories of LV myocardial damage, subepicardial edema, and hyperemia were significantly (respectively *p* = 0.0395 and *p* = 0.0011) less frequent in group 1 than in group 2; subepicardial LGE was significantly (*p* = 0.00023) less frequent in group 1 than group 2, while intramural LGE was significantly (*p* = 1.02 × 10^−4^) more frequent in group 1 than group 2. LV myocardial edema was significantly predominant in the septal location (*p* = 0.0076) and significantly less extensive in the inferior and lateral locations (respectively, *p* = 0.0450 and *p* = 0.0405) in group 1 than in group 2. LV myocardial hyperemia was significantly more prevalent in septal (*p* = 0.0240) and apical locations (*p* = 0.0440) in group 1 than in group 2. LV myocardial LGE was significantly less extensive in the anterior (*p* = 0.0244), lateral (*p* = 6.9 × 10^−4^), and middle (*p* = 0.0076) locations in group 1 than in group 2. Among LV-affected segments, edema was more frequently observed (*p* = 0.0035) in septal location and less frequently observed (*p* = 0.0308) in lateral location in group 1 than group 2; hyperemia was less frequently observed (*p* = 0.0308) in the mid-cavity (*p* = 0.0457) in group 1 than group 2; LGE was more frequently observed (*p* = 0. 0078) in septal location in group 1 than group 2, with a major extent in the mid-cavity in group 2 (*p* = 0.0076) (Figure 2, Figure 3 and Figure 4).

## 4. Discussion

In our study, as in most literature studies, the binary classification between “infarct-like” and “non-infarct-like” AM was used, based on the presence/absence of clinical suspicion of ACS and the consequent need for a prompt search for coronary stenosis. As in our case, in the work of Chopra et al. and Schwab et al., the diagnosis of AM was based on the CMR presentation in accordance with “Lake Louise” criteria and on clinical data [2,18].

In 78.6% of group 1 patients and 72% of group 2 patients, 3/3 LLC were achieved, whereas in the remaining cases, 2/3 of the aforementioned criteria were achieved; however, in all patients, the diagnosis of AM was clinically confirmed. These percentages are influenced, of course, by the retrospective nature of the study and the selection criteria of the CMR examinations; in fact, examinations of patients with a diagnosis or clinical suspicion of AM that did not meet at least two of the three diagnostic criteria in CMR were excluded. However, in both study samples, the most frequently missed parameter was hyperemia (two cases in group 1, five in group 2); this is in agreement with the known “weakness” of hyperemia as a diagnostic criterion in CMR compared with the other two (edema and LGE) [1,13,15,16,17,18,19,20,21].

### 4.1. “Infarct-like” Presentation

In this study, the “infarct-like” clinical presentation was observed in 35.9% of patients with AM confirmed by CMR, according to the original LLC. This percentage differs from those reported by Lurz et al. (52.8%) and by Chopra et al. (54.5%), but it is similar to that reported by Francone et al. (36.8%), although the classification and inclusion criteria are partly different [1,2,22]. In the literature, it is reported that the clinical presentation depends on the virus responsible for AM etiology. The “non-infarct-like” pattern is observed in the case of Parvovirus B19 infection; the virus determines polyserositis and, therefore, pericarditis after initial viremia with subepicardial localization [3,19,22,23]. Considering the direct contact of the infero-lateral wall of the LV with the pericardium, this represents the main site of damage by continuity diffusion of viral pathogens.

When myocardial involvement predominates in the epicardial region, this may justify a lesser severity of the clinical presentation and the presence of symptoms related to the involvement of the serosa, conditioning the clinical suspicion of myocarditis.

In the “infarct-like” pattern, the interventricular septum is usually involved due to the neurotropism of some viruses, such as Human Herpesvirus 6 (HHV6), which has a predilection to localize in myocardial conduction tissue. When the damage involves and predominates within the thickness of the myocardial wall, in particular the mesocardial portion, the clinical presentation may simulate that of an ACS [19].

Finally, the type of clinical presentation is also described as being associated with different values of diagnostic sensitivity of CMR examinations, with higher values in cases of “infarct-like” AM, correlating with the greater extent of myocardial damage and interstitial enhancement [1].

### 4.2. Morphological and Functional Parameters

In the majority (>70%) of patients, LV size, myocardial thickness, and function were substantially preserved, and no significant differences were observed between the two study groups [1,19,24].

Nevertheless, in group 1, the EF was lower than that of the patients in group 2. In contrast, the studies by Schwab et al. and Chopra et al. reported, respectively, a significant incidence of reduced EF in patients with “infarct-like” AM, and a lower EF in patients with “infarct-like” AM compared with other types of AM [2,18]. The failure to find frequent and significant impairment of the LV systolic function finds justification in the focality of the myocardial process, with sparing of the subendocardial side, such that EF is not significantly altered [19,24]. In our study, a severe reduction in FE was found in 2/14 of group 1 and 3/25 of group 2 patients, respectively; in 1/2 of group 1 and 2/3 of group 2 patients, this functional impairment was associated with extensive signal alterations in the sequences of tissue characterization. In particular, in group 2, EF showed a significant (*p* = 0.002) inverse correlation with the extension of the LGE, which was significantly greater than in group 1 (*p* = 0.0041). It should be noted, however, that the extent of LGE tends to overestimate myocardial damage because, in the acute phase, not all myocardiocytes in the areas of enhancement are damaged [19].

Pericardial effusion was present, predominantly affecting the lateral segments, in both study samples (42.9% group 1; 40% group 2), in agreement with the prevalence ranging from 32% to 57% described by several case reports. Pericardial effusion is a frequently encountered finding, although not specific to myocarditis, and its presence is indicative of possible involvement of the serosa [16].

### 4.3. Tissue Characterization Parameters

CMR’s ability to detect alterations in signal intensity is closely related to the presence of the three elements that characterize myocardial inflammation: edema, hyperemia, and fibrosis/necrosis [1].

LLC represents the standard for CMR diagnosis of AM and takes into account the three markers of myocardial injury, namely, intracellular and interstitial edema on T2-weighted imaging, hyperemia and capillary leakage with EGE, and necrosis and fibrosis with LGE. The new LLC, updated in 2018, redefined imaging diagnosis according to the combined presence of a T1 criterion (presence of LGE or increased T1 mapping or extracellular volume values) and a T2 criterion (hyperintensity in T2 weighted STIR or increased T2 mapping values). These changes have significantly improved both the specificity and diagnostic accuracy of the LLC, especially in patients who do not present with infarct-like symptoms [25]. Unfortunately, in our hospital, the CMR mapping technique was not available at the time of the study.

Usually, tissue alterations (edema, hyperemia, and fibrosis/necrosis) are localized in the thickness of the ventricular wall as linear or patchy areas of hyperintensity with a subepicardial or intramural distribution, as observed in our study [10,16,19,20,24].

For both forms of AM, we observed that the subepicardial pattern was the predominant one, followed by the intramural pattern, for each tissue characterization parameter, except for LGE in group 1, in which mesocardial involvement prevailed. Furthermore, subepicardial involvement in non-infarct-like patients was found to be significantly more frequent than that observed in “infarct-like” patients for all parameters (edema, *p* = 0.0395; hyperemia, *p* = 0.0011; LGE, *p* = 0.00023), denoting a preferential subepicardial involvement of cellular damage in “non infarct-like” AM. In patients with “infarct-like” AM, on the other hand, hyperemia and LGE presented prevalent mesocardial involvement, and mesocardial LGE in the “infarct-like forms” was significantly more frequent than that observed in “non-infarct-like” AM. These data can be interpreted as a tendency for all three parameters of myocardial damage to be localized to the subepicardial level in “non-infarct-like” AM, whereas for the “infarct-like” forms, the tendency, especially for hyperemia and LGE, is to mesocardial involvement of the ventricular wall.

The mechanisms underlying this predilection are not known; however, it is hypothesized that, when the damage involves and predominates within the thickness of the myocardial wall, the clinical presentation is more likely to resemble or mimic that of an ACS.

In fact, the AMI, is frequent involvement in the intermediate portion of the parietal thickness, starting from the subendocardial layer towards the subepicardial one it may also become transmural, so that, when there is involvement of the middle average parietal thickness, even in the absence of segmental distribution of vascular distribution and subendocardial sparing, the resulting clinical presentation may induce the doubt that it is an ACS.

On the contrary, when the myocardial involvement predominates only on the epicardial side, this could justify a minor severity of the clinical presentation with the presence of symptoms related to serosal involvement, thus conditioning a more targeted suspicion towards a condition of myocardial-pericarditis.

In particular, it has been reported that in AM with viral etiology, the subepicardial localization may result from the ability of cardiotropic viruses to cause polyserositis and, therefore, pericarditis after initial viremia. Given the direct contact of the infero-lateral wall of the LV with the pericardium, this represents the main site of damage by diffusion by continuity of viral pathogens (especially PVB19) [19].

This mechanism is also able to justify, in part, the different locations of the signal alterations found in the two study samples; the “non-infarct-like” form showed, in our study, a significantly greater distribution of edema (*p* = 0.0308) and a significantly greater extent of edema and LGE at the level of the lateral (*p* = 0.0405 and *p* = 6.91 × 10^−4^, respectively) and inferior segments (edema *p* = 0.045) of the LV, particularly in the mid-segment (*p* = 0.0076). The infero-lateral middle (basal) wall of the LV is, in fact, described as the site commonly affected by the phlogistic process, and it is the one in direct contact with the pericardium, accounting for its frequent involvement in AM caused by cardiotropic viruses (PVB19), as mentioned previously [2,5,12,21,25,26]. This localization of tissue damage in relation to PVB19 infection was described by Mahrholdt et al. in association with an “infarct-like” presentation of AM [19]. PVB19 has a predilection for endothelial cells, which may lead to endothelial dysfunction and vasospasm, causing anginal symptoms with typical infarct-like ECG changes [19].

In contrast, in this study, the “infarct-like” form showed a significantly greater distribution of edema and LGE (*p* = 0.0035 and *p* = 0.0078, respectively) and a significantly greater extent of edema and hyperemia (respectively, *p* = 0.0076 and *p* = 0.024) at the level of the septal segments of the LV.

Mahrholdt et al. [19] described the presence of LGE at the interventricular septum as being associated with HHV6 infection. The involvement of this site is related to the neurotropism of the virus and to its localization in the myocardial conduction tissue, conditioning a clinical presentation characterized by heart failure, arrhythmias, and branch blocks.

Moreover, the finding of LGE in the interventricular septum at presentation is one of the strongest predictors of future chronic ventricular dysfunction and dilatation, although the underlying mechanisms remain unknown [16]. Thus, unlike reported by Mahrholdt et al., in our study, the localization of tissue damage in “infarct-like” AM was found in the midparietal location in the interventricular septum, whereas the localization of signal alterations in patients with “non-infarct-like” presentation was found in the subepicardial location at the LV free wall [19].

The mechanisms underlying the different localization of tissue damage between the two forms of clinical presentation of AM are not known to us, also due to the absence of biopsy findings and the lack of viral serology assessment.

However, despite the lack of information about the etiological agent, it seems unlikely that the viral type alone could be sufficient to explain our results. In particular, PVB19, described as responsible for infarct-like forms, is localized at the level of the free wall of the LV, whereas, in our study, tissue damage in “infarct-like” forms was localized at the level of the septal segments [19].

### 4.4. Study Limitations

This study has some limitations, many of which are related to its retrospective nature.

First, a possible selection “bias” of the sample could have resulted from having considered only CMR examinations that satisfied LLC, excluding other even more severe conditions. For this reason, we did not conduct an analysis of the sensitivity, specificity, and predictive value of the parameters we evaluated.

Moreover, no CMR follow-up examinations were considered in order to investigate the prognostic implications of the obtained results.

The sample size of the study is small, reflecting the activity of a single hospital center; however, it is of the order of magnitude overlapping with that of some recent studies [1,15,25]. The CMR diagnosis was confirmed by clinical diagnosis in all enrolled patients, however, without biopsy. Not all group 2 patients underwent a coronary circulation study to exclude coronary artery disease, and their cardiovascular risk factors were not considered.

It is possible that in group 2, there were patients with “infarct-like” symptoms according to the classical description (chest pain, ST-segment elevation, elevation of troponins) without, however, a clinical presentation that would justify a coronarography in the suspicion of an ACS. It is also possible that referral for coronarography reflected the operator-dependent nature of clinical evaluation.

Although the Perfetti et al. method has not yet been validated, it appears to be effective in identifying areas of hyperemia, overcoming many of the limitations of the T1 FSE sequences, and we employed it in this retrospective study (in the absence of classical FSE acquisitions), even if the evaluation was only qualitative [10,15].

Finally, a quantitative analysis (such as “mapping”) was not performed for all the CMR parameters considered, resorting to qualitative/semiquantitative assessments for the tissue characterization parameters [26].

## 5. Conclusions

In patients with a clinical presentation characterized by acute chest pain and ECG alterations accompanied by elevation of myocardiocyte necrosis indices without angiographic evidence of significant coronary stenosis, CMR plays an essential role in the diagnosis of alternative conditions with a similar infarct-like presentation, primarily AM. Our results show that there are differences in the localization and distribution of myocardial tissue damage assessed by CMR among forms of AM accompanied by an “infarct-like” clinical presentation, such as to condition the execution of an emergency coronarography in the suspicion of an ACS, compared with other forms of AM with a “non-infarct-like” presentation. In contrast, the morphological and functional parameters do not seem to be affected by the different clinical presentations. However, the mechanisms underlying the different preferential sites of damage observed in our study between the two different clinical forms of AM are not known, and we do not exclude the possibility of their prognostic implications. To date, there is no other study in the literature that has related the frequency and location of the parameters investigated by us with AM clinical presentation, and, therefore, further studies, more extensive and multicenter, are necessary so that we can validate, justify, or refute our results in the light of future scientific evidence, also applying quantitative techniques (such as “mapping”) for tissue characterization.

## Figures and Tables

**Figure 1 diagnostics-13-02498-f001:**
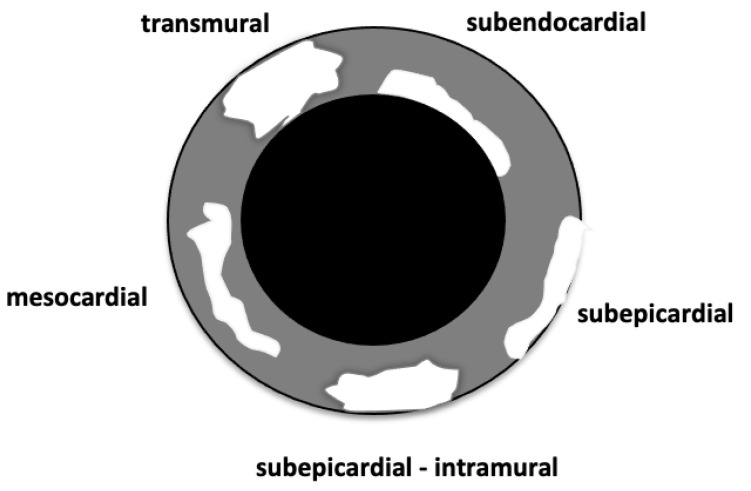
Schematic representation of left ventricle wall (grey colored) on short-axis view; white shapes represent the pattern of myocardial involvement according to the location of the alteration within the myocardial thickness; Ventricular cavity in black.

**Figure 2 diagnostics-13-02498-f002:**
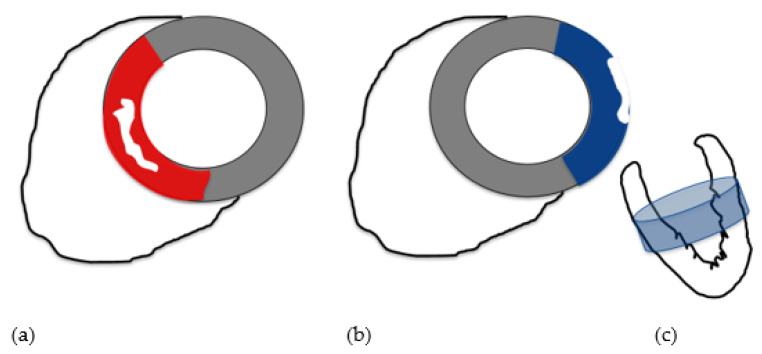
In group 1 (**a**), CMR tissue characterization parameters were mainly altered in septal location (in red) with mesocardial myocardial involvement (white shape); in group 2 (**b**), CMR tissue characterization parameters were mainly altered in lateral location (in blue) with subepicardial myocardial involvement (white shape) mainly at the mid-cavity (in light blue, (**c**)).

**Figure 3 diagnostics-13-02498-f003:**
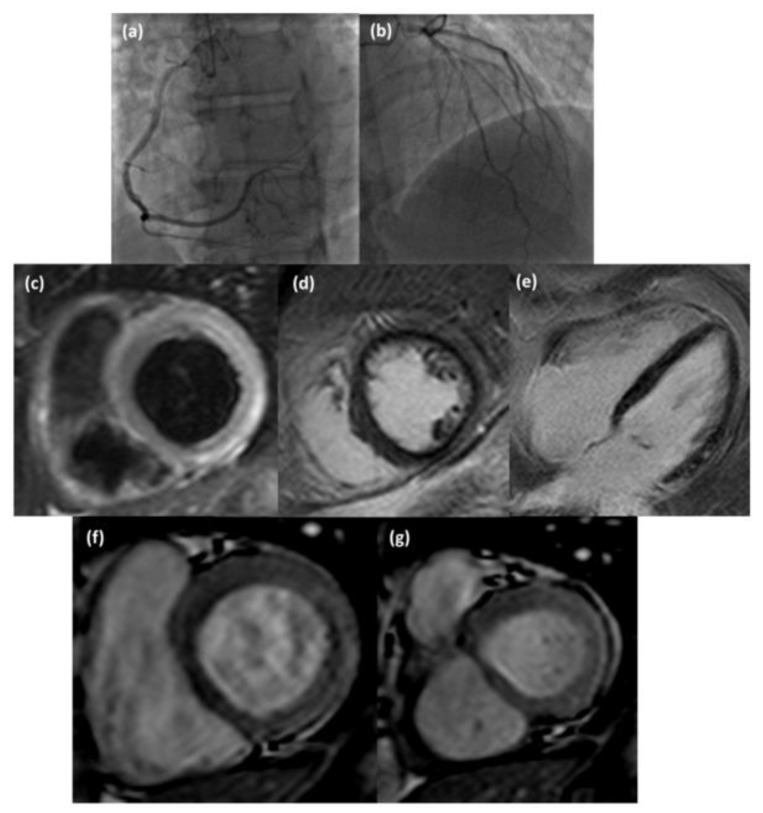
Coronary angiography images (**a**,**b**) of a 42-year-old man suspected of acute coronary syndrome that did not show hemodynamically significant stenosis; CMR examination revealed the presence of subepicardial edema (T2-weighted STIR short axis view, (**c**)), subepicardial late gadolinium enhancement (inversion recovery T1-weighted short axis view, (**d**) and four chamber view, (**e**)) and hyperemia (cine steady-state free precession short axis view in diastol (**f**) and systole (**g**)) at the basal inferior-lateral wall of the LV.

**Figure 4 diagnostics-13-02498-f004:**
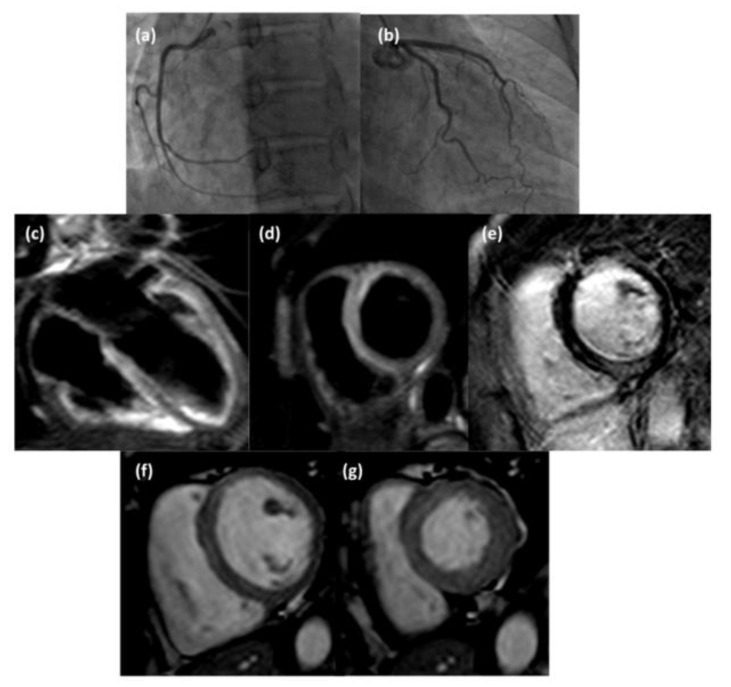
Coronary angiography images (**a**,**b**) of a 48-year-old woman suspected of acute coronary syndrome that did not show hemodynamically significant stenosis; CMR examination revealed the presence of subepicardial-intramural edema (T2-weighted STIR 4-chamber view (**c**) and short axis view (**d**)) at the basal septum, intramural late gadolinium enhancement (inversion recovery T1-weighted short axis view, (**e**)) at the inferior septal wall, and slight mesocardial diastolic hyperemia at interventricular septum (cine steady-state free precession short axis view in diastol (**f**) and systole (**g**)).

**Table 1 diagnostics-13-02498-t001:** Baseline Patients’ Characteristics.

Parameter	Group 1 % (*n*)	Group 2 % (*n*)	Group 1 vs. Group 2 *p*-Value (Test)
Patients (*n*)	14	25	-
Age (years: mean ± SD)	48.15 ± 19.13	36.92 ± 14.43	**0.0014 (T)**
Gender			0.50 (C)
Males	57.14 (8)	68.00 (17)	
Females	42.86 (6)	32.00 (8)	
Fever	14.29 (2)	36.00 (9)	0.266 (F)
Chest pain	100.0 (14)	64.00 (16)	**0.0149 (F)**
Dyspnea	35.71 (5)	64.00 (16)	0.089 (C)
Palpitations	21.43 (3)	20.00 (5)	1.00 (F)
ECG changes	100.0 (14)	68.00 (17)	**0.0337 (F)**
ST segment elevation	64.29 (9)	0.00 (0)	**<0.0001 (F)**
ST segment depression	35.71 (5)	0.00 (0)	**0.00369 (F)**
T-wave inversion	7.14 (1)	0.00 (0)	0.463 (F)
Bundle branch bloc	7.14 (1)	0.00 (0)	0.463 (F)
Non-specific ST segment change	0.00 (0)	48.00 (12)	**0.00259 (F)**
Non-specific T wave changes	0.00 (0)	20.00 (5)	0.139 (F)
Serum marker	100.0 (14)	92.00 (23)	0.528 (F)
CK-MB	92.86 (13)	92.00 (23)	1.00 (F)
Tn-I	100.0 (14)	20.00 (5)	**<0.0001 (C)**
Pharyngitis/upper respiratory tract	7.14 (1)	12.00 (3)	1.00 (F)
Colitis/gastroenteritis	7.14 (1)	20.00 (5)	0.391 (F)
Influenza syndrome	7.14 (1)	0.00 (0)	0.463 (F)
Flu	0.00 (0)	32.00 (8)	**0.0337 (F)**
Serological tests	14.29 (2)	0.00 (0)	0.144 (C)
IgM for Coxsackie B	7.14 (1)	0.00 (0)	0.463 (F)
IgM for HSV1	7.14 (1)	0.00 (0)	0.463 (F)
Throat swab	0.00 (0)	8.00 (2)	0.528 (F)
Influenza virus	0.00 (0)	8.00 (2)	0.528 (F)
Time to CMR (days: mean ± SD)	2.89 ± 3.19	5.28 ± 6.23	0.190 (T)
Duration of hospital stay (days: mean ± SD)	7.40 ± 5.14	11.10 ± 10.51	0.226 (T)
Duration of symptoms (days: mean ± SD)	12.30 ± 4.11	15.70 ± 2.32	**0.0020 (T)**
Time to follow-up CMR (months: mean ± SD)	5.40 ± 1.20	6.20 ± 1.10	**0.0417 (T)**

Baseline characteristics of groups 1 and 2. In bold, the *p*-values are reported as statistically significant. T = Student *T*-test; C = Chi-square test with Yates correction; F = Fisher’s exact test.

**Table 2 diagnostics-13-02498-t002:** CMR Morpho-Functional Parameters.

Parameter	Group 1 (Mean ± DS)	Group 2 (Mean ± DS)		*p*-Value
EF %	56.64 ± 11.79	58.52 ± 11.81		0.323
EDV (mL/m^2^)	80.79 ± 20.32	76.24 ± 14.41		0.217
ESV (mL/m^2^)	35.79 ± 13.18	31.76 ± 12.32		0.184
ESV (mL/m^2^)	47.21 ± 15.87	44.28 ± 11.68		0.263
Mass (g/m^2^)	63.92 ± 18.85	63.20 ± 19.25		0.455
			Statistical test	
	Group 1	Group 2	Hypothesis	*p*-value (Test)
% contractility deficit	(mean ± DS)	(mean ± DS)		
Segmentary	14.28 (2/14)	12.00 (3/25)	14.28 > 12.00	0.364 (F)
Diffuse	14.28 (2/14)	8.00 (2/25)	14.28 > 8.00	0.332 (F)
Morphological parameters	(mean ± DS)	(mean ± DS)		
% of patient with LV dilatation	7.14 (1/14)	16.00 (4/25)	7.14% < 16.00	0.308 (F)
Mean	59.00 ± 0.00	60.50 ± 0.50	59.00 < 60.50	0.167 (T)
% of patients with LV thickening	7.14 (1/14)	8.00 (2/25)	7.14 < 8.00	0.460 (F)
Mean	14.00 ± 0.00	15.00 ± 1.00	14.00 < 15.00	0.333 (T)
T = Student *T*-test; F = Fisher’s exact test				
Correlations	Pearson’s linear correlation coefficient			*p*-value
G2: EF vs. LGE	**−0.588**			**0.0020**
G1: EF vs. LGE	−0.328			0.252
	PERICARDIAL EFFUSION			
	Group 1	Group 2	Hypothesis	*p*-value
% Patients	42.86 (6/14)	40.00 (10/25)	42.86 > 40.00	0.434
% Segments of LV	16.81 (40/238)	11.53 (49/425)	16.81 > 11.33	**0.0365**
N.Patients/N. Segments	35.00 (14/40)	51.02 (25/49)	35.00 < 51.02	0.0967

Results of CMR morpho-functional parameter analysis; in bold are reported “*p*-values” statistically significant. LV: left ventricle; EF: ejection fraction; EDV: end diastolic volume; ESV: end systolic volume; LGE: late gadolinium enhancement.

**Table 3 diagnostics-13-02498-t003:** CMR Tissue Characterization Findings.

	Group 1			Group 2		
	Edema	Hyperemia	LGE	Edema	Hyperemia	LGE
% of patients	100%	85.71%	92.86%	96.00%	80.00%	96.00%
	(14/14)	(12/14)	(13/14)	(24/25)	(20/25)	(24/25)
% of affected segments	20.59%	13.86%	16.81%	22.12%	10.59%	26.12%
	(49/238)	(33/238)	(40/238)	(94/425)	(45/425)	(111/425)
Pattern						
Subepicardial	48.98%	57.58%	17.50%	47.97%	68.89%	54.95%
	(24/49)	(19/33)	(7/40)	(45/94)	(31/45)	(61/111)
Mesocardial	14.29%	33.33%	25.00%	7.45%	4.44%	4.50%
	(7/49)	(11/33)	(10/40)	(7/94)	(2/45)	(5/111)
Transmural	10.20%	0.00%	2.50%	5.32%	0.00%	0.90%
	(5/49)	(0/33)	(1/40)	(5/94)	(0/45)	(1/111)
Subendocardial	0.00%	0.00%	0.00%	0.00%	0.00%	0.00%
	(0/49)	(0/33)	(0/40)	(0/94)	(0/45)	(0/111)
Subepicardial with intramural extension	26.53%	9.09%	55.00%	39.36%	26.67%	39.64%
	(13/49)	(3/33)	(22/40)	(37/94)	(12/45)	(44/111)
Extent						
Anterior	19.05%	19.05%	4.76%	13.33%	8.00%	20.00%
	(8/42)	(8/42)	(2/42)	(10/75)	(6/75)	(15/75)
Septal	20.00%	8.57%	17.14%	7.20%	1.60%	10.40%
	(14/70)	(6/70)	(12/70)	(9/125)	(2/125)	(13/125)
Inferior	10.05%	4.76%	19.05%	33.33%	9.33%	30.67%
	(8/42)	(2/42)	(8/42)	(25/75)	(7/75)	(23/75)
Lateral	25.71%	24.29%	22.86%	39.20%	24.00%	47.20%
	(18/70)	(17/70)	(16/70)	(49/125)	(30/125)	(59/125)
Apex	7.14%	0.00%	14.28%	4.00%	0.00%	4.00%
	(1/14)	(0/14)	(2/14)	(1/25)	(0/25)	(1/25)
Basal	27.38%	17.86%	22.62%	32.67%	11.33%	30.00%
	(23/84)	(15/84)	(19/84)	(49/150)	(17/150)	(45/150)
Middle	17.86%	14.29%	14.29%	18.00%	16.67%	29.33%
	(15/84)	(12/84)	(12/84)	(27/150)	(25/150)	(44/150)
Apical	17.86%	10.71%	12.50%	17.00%	3.00%	21.00%
	(10/56)	(6/56)	(7/56)	(17/100)	(3/100)	(21/100)
Apex	7.14%	0.00%	14.28%	4.00%	0.00%	4.00%
	(1/14)	(0/14)	(2/14)	(1/25)	(0/25)	(1/25)
Distribution						
Anterior	16.32%	24.24%	5.00%	10.64%	13.33%	13.51%
	(8/49)	(8/33)	(2/40)	(10/94)	(6/45)	(15/111)
Septal	28.57%	18.18%	30.00%	9.57%	4.44%	11.71%
	(14/49)	(6/33)	(12/40)	(9/94)	(2/45)	(13/111)
Inferior	16.32%	6.06%	20.00%	26.59%	15.55%	20.72%
	(8/49)	(2/33)	(8/40)	(25/94)	(7/45)	(23/111)
Lateral	36.73%	51.51%	40.00%	52.13%	66.67%	53.15%
	(18/49)	(17/33)	(16/40)	(49/94)	(30/45)	(59/111)
Apical	2.04%	0.00%	5.00%	1.06%	0.00%	0.90%
	(1/49)	(0/33)	(2/40)	(1/94)	(0/45)	(1/111)
Basal	46.94%	45.46%	47.50%	52.13	37.78%	40.54%
	(23/49)	(15/33)	(19/40)	(49/94)	(17/45)	(45/111)
Middle	30.61%	36.36%	30.00%	28.72%	55.55%	39.64%
	(15/49)	(12/33)	(12/40)	(27/94)	(25/45)	(44/111)
Apical	20.41%	18.18%	17.50%	18.08%	6.67%	18.92%
	(10/49)	(6/33)	(7/40)	(17/94)	(3/45)	(21/111)
Apex	2.04%	0.00%	5.00%	1.06%	0.00%	0.90%
	(1/49)	(0/33)	(2/40)	(1/94)	(0/45)	(1/111)

Results of CMR tissue characterization assessment.

**Table 4 diagnostics-13-02498-t004:** CMR Tissue Characterization Parameters Analysis.

	Edema: Group 1 vs. Group 2		Hyperemia: Group 1 vs. Group 2		LGE: Group 1 vs. Group 2	
Parameter	Hypothesis	*p*-Value	Hypothesis	*p*-Value	Hypothesis	*p*-Value
% of patients	100.00 > 96.00	0.641 (F)	85.71 > 80.00	0.314 (F)	92.86 < 96.00	0.472 (F)
% of segments	20.59 < 22.12	0.359 (C)	13.86 > 10.59	0.129 (C)	16.81 < 26.12	**0.0041 (C)**
% Pattern						
Subepicardial	48.98 > 47.97	0.480 (C)	57.58 < 68.89	0.215 (C)	17.50 < 54.95	**4.87 × 10^−5^ (C)**
Mesocardial	14.29 > 7.45	0.156 (C)	33.33 > 4.44	**1.05 × 10^−3^ (C)**	25.00 > 4.50	**3.28 × 10^−4^ (C)**
Transmural	10.20 > 5.32	0.229 (C)	0.00 = 0.00	0.00 (F)	2.50 > 0.90	0.481 (C)
Subendocardial	0.00 = 0.00	0.00 (C)	0.00 = 0.00	0.00 (F)	0.00 = 0.00	0.00 (F)
Subepicardial and intramural	26.53 < 39.36	**0.0467 (F)**	9.09 < 26.67	**0.0359 (F)**	55.00 > 39.64	**0.0366 (F)**
% Pattern						
Subepicardial	75.51 < 87.23	**0.0395 (F)**	66.67 < 95.56	**0.0011 (C)**	72.50 < 94.59	**0.00023 (C)**
Mesocardial	40.82 > 46.81	0.306 (C)	42.24 > 31.11	0.215 (C)	80.00 > 44.14	**1.02 × 10^−4^ (C)**
Transmural	10.20 > 5.32	0.229 (C)	0.00 = 0.00	0.00 (F)	2.50 > 0.90	0.481 (C)
Subendocardial	0.00 = 0.00	0.00 (C)	0.00 = 0.00	0.00 (F)	0.00 = 0.00	0.00 (F)
Extent						
Anterior	19.05 > 13.33	0.290 (C)	19.05 > 8.00	0.071 (C)	4.76 < 20.00	**0.0244 (C)**
Septal	20.00 > 7.20	**0.0076 (C)**	8.57 > 1.60	**0.0240 (C)**	17.14 > 10.40	0.130 (C)
Inferior	10.05 < 33.33	**0.0450 (F)**	4.76 < 9.33	0.299 (C)	19.05 < 30.67	0.126 (C)
Lateral	25.71 < 39.20	**0.0405 (C)**	24.29 > 24.00	0.448 (C)	22.86 < 47.20	**6.91 × 10^−4^ (C)**
Apex	7.14 > 4.00	0.371 (C)	0.00 = 0.00	0.00 (F)	14.28 > 4.00	0.298 (C)
Basal	27.38 > 32.67	0.244 (C)	17.86 > 11.33	0.116 (C)	22.62 < 30.00	0.144 (C)
Middle	17.86 < 18.00	0.440 (C)	14.29 < 16.67	0.385 (C)	14.29 < 29.33	**0.0076 (C)**
Apical	17.86 > 17.00	0.466 (C)	10.71 > 3.00	**0.0440 (F)**	12.50 < 21.00	0.134 (C)
Apex	7.14 > 4.00	0.371 (C)	0.00 = 0.00	0.00 (F)	14.28 > 4.00	0.298 (C)
Distribution						
Anterior	16.32 > 10.64	0.240 (C)	24.24 > 13.33	0.173 (C)	5.00 < 13.51	0.121 (C)
Septal	28.57 > 9.57	**0.0035 (C)**	18.18 > 4.44	0.0817 (F)	30.00 > 11.71	**0.0078 (C)**
Inferior	16.32 < 26.59	0.120 (C)	6.06 < 15.55	0.174 (C)	20.00 > 20.72	0.448 (C)
Lateral	36.73 < 52.13	**0.0308 (F)**	51.51 < 66.67	0.132 (C)	40.00 < 53.15	0.107 (C)
Apex	2.04 > 1.06	0.454 (F)	0.00 = 0.00	0.00 (F)	5.00 > 0.90	0.154 (F)
Basal	46.94 < 52.13	0.340 (C)	45.36 > 37.78	0.327 (C)	47.50 > 40.54	0.282 (C)
Middle	30.61 > 28.72	0.483 (C)	36.36 < 55.55	**0.0457 (F)**	30.00 < 39.64	0.186 (C)
Apical	20.41 > 18.08	0.456 (C)	18.18 < 6.67	0.112 (C)	17.50 < 18.92	0.484 (C)
Apex	2.04 > 1.06	0.454 (F)	0.00 = 0.00	0.00 (F)	5.00 > 0.90	0.154 (F)

Results of CMR tissue characterization parameter analysis; in bold are reported “*p*-values” statistically significant. LGE: late gadolinium enhancement. C = Chi-square test with Yates correction; F = Fisher’s exact test.

## Data Availability

Data is contained within the article.

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
