# Peer review of "Infarct-like versus Non-Infarct-like Clinical Presentation of Acute Myocarditis: Comparison of Cardiac Magnetic Resonance (CMR) Findings"

_diagnostics, 2023, doi:10.3390/diagnostics13152498_

Round 1

Reviewer 1 Report

The authors provide a description of the distinct manifestations of two types of myocarditis, namely "infarct-like" and "non-infarct-like," which were retrospectively observed using CMR.

First, it is worth considering whether it is standard practice for patients with myocarditis to undergo CMR. Typically, patients with myocarditis experience symptoms such as tachycardia, dyspnea, and hypotension. Conducting a CMR examination can be time-consuming and requires the patient's cooperation and ability to breathe properly. Regarding the timing of CMR, the study conducted CMR scans 2.89 days (Group 1) and 5.28 days (Group 2) after diagnosis. Is there a correlation between the number of days and the clinical severity of the patients? Additionally, does the CMR examination exclude patients with more severe conditions? If this is the case, it should be mentioned as a limitation. If patients with more severe conditions were not excluded, it would be helpful to describe to what extent the clinical unit schedules the patient for a CMR scan once they have recovered.

Second, the author emphasizes in the discussion that different viral infections may result in distinct CMR manifestations. Since this study is a single-center retrospective analysis, it should not be difficult to determine the viral infection status of the myocarditis patients. Therefore, it is necessary to provide information on the viral infection status of the patients to investigate if there are genuine differences between Group 1 and Group 2.

Third, there is no detailed explanation in the full text for the abbreviation "FE" mentioned in line 322 and Table 1. Could the author be referring to "EF" (ejection fraction)?

Fourth, there is an error in line 184 where "5,28 ± 6,23 days" has incorrect punctuation.

Finally, it is recommended to reorganize the paragraph from line 219 to 230. The sentence breaks and punctuation are unclear, and there are unknown symbols, making it difficult to comprehend.

Author Response

First of all, we would like to thank the reviewer for his comments that improve the scientific quality of this work.

Our reply are reported in red color within the text.

The authors provide a description of the distinct manifestations of two types of myocarditis, namely "infarct-like" and "non-infarct-like," which were retrospectively observed using CMR.

First, it is worth considering whether it is standard practice for patients with myocarditis to undergo CMR. Typically, patients with myocarditis experience symptoms such as tachycardia, dyspnea, and hypotension. Conducting a CMR examination can be time-consuming and requires the patient's cooperation and ability to breathe properly. Regarding the timing of CMR, the study conducted CMR scans 2.89 days (Group 1) and 5.28 days (Group 2) after diagnosis. Is there a correlation between the number of days and the clinical severity of the patients?

It may be that patients underwent to coronary angiography had a more sever clinical presentation but clinical symptoms were not scored (using a severity scale) to evaluate the gravity. Therefore it is not possible to asses this correlation.

Additionally, does the CMR examination exclude patients with more severe conditions? If this is the case, it should be mentioned as a limitation.

Patients were retrospectively selected by CMR diagnosis of myocarditis so other even more sever conditions were not included. This aspect has been added in limitations as you suggested.

If patients with more severe conditions were not excluded, it would be helpful to describe to what extent the clinical unit schedules the patient for a CMR scan once they have recovered.

Second, the author emphasizes in the discussion that different viral infections may result in distinct CMR manifestations. Since this study is a single-center retrospective analysis, it should not be difficult to determine the viral infection status of the myocarditis patients. Therefore, it is necessary to provide information on the viral infection status of the patients to investigate if there are genuine differences between Group 1 and Group 2.

Added in Results, 3.1. Patients paragraph.

Third, there is no detailed explanation in the full text for the abbreviation "FE" mentioned in line 322 and Table 1. Could the author be referring to "EF" (ejection fraction)?

Yes, ejection fraction. Corrected throughout the text.

Fourth, there is an error in line 184 where "5,28 ± 6,23 days" has incorrect punctuation.

Corrected.

Finally, it is recommended to reorganize the paragraph from line 219 to 230. The sentence breaks and punctuation are unclear, and there are unknown symbols, making it difficult to comprehend.

Corrected.

Reviewer 2 Report

I was intrigued by the title of the manuscript assuming that infarct-like myocarditis refers to LGE pattern on CMR. However, as I have quickly realized it refers to clinical presentation instead. Please clarify this issue, maybe changing the wording to "anginal symptoms".  I have several comments:

1. Due to retrospective nature of the study there is a potentially strong inclusion bias. Patients with infarct-like characteristic - chest pain, ECG changes and elevated cardiac markers were present in both groups, but the only difference was coronary angiogram (CA) performed in the 1st group. It is possible that referral for CA depended on the physician on duty not the presentation itself? Please discuss or include in the limitations section. For these reason I also suggest to include Table 1 with baseline characteristic to see what the true difference between both groups was. 

2. The study group is small, did it include Covid-19 related AM? Was it excluded? Please add this information to text. 

3. Why were modified LLC not used? They were shown to have better diagnostic accuracy than original (old ones). Please discuss. 

4. I think the idea behind parvovirus-19 relation to infarct-like presentation is hypothetically explained by its predilection to endothelial cells, which may lead to endothelial dysfunction and vasospasm causing anginal symptoms with typical infarct-like ECG changes (Marholdt H et la. Circulation 2006)

5. What are the potential clinical consequences of your results? Would it stop physicians from performing unnecessary angiograms? Probably not. Were there any differences in clinical follow-up? Duration of hospital stay, duration of symptoms, cardiac events? Please include in your analysis as the study is retrospective. This would add clinical value to your study.  

6. Finally - English needs improvement. For example Line 68 - we instead of were...etc. 

None 

Author Response

First of all, we would like to thank the reviewer for his comments that improve the scientific quality of this work.

Our reply are reported in red color within the text.

I was intrigued by the title of the manuscript assuming that infarct-like myocarditis refers to LGE pattern on CMR. However, as I have quickly realized it refers to clinical presentation instead. Please clarify this issue, maybe changing the wording to "anginal symptoms".  The expression "infarct-like" is widely used in literature to concern to the clinical manifestation of acute myocarditis (Chopra, Francone, Faletti, Schwab...). To avoid misunderstanding we specified in the title "clinical presentation of acute myocarditis" while no reference to LGE is made.

I have several comments:

  1. Due to retrospective nature of the study there is a potentially strong inclusion bias. Patients with infarct-like characteristic - chest pain, ECG changes and elevated cardiac markers were present in both groups, but the only difference was coronary angiogram (CA) performed in the 1st group. It is possible that referral for CA depended on the physician on duty not the presentation itself?  Please discuss or include in the limitations section. For these reason I also suggest to include Table 1 with baseline characteristic to see what the true difference between both groups was. Certainly the clinical evaluation has played an important role in the definition of the diagnostic procedure. Added in study limitations. Added the new table 1 with baseline characteristic of the study groups.

2. The study group is small, did it include Covid-19 related AM? Was it excluded? Please add this information to text. Added in paragraph 2.2.Patients; this study was conducted before the outbreak of the COVID-19 pandemic.

3. Why were modified LLC not used? They were shown to have better diagnostic accuracy than original (old ones). Please discuss. New LLC redefined imaging diagnosis according to the combined presence of a T1 criterion (presence of LGE or increased T1 mapping or extracellular volume values) and a T2 criterion (hyperintensity in T2 weighted STIR or increased T2 mapping values). Unfortunately in our hospital CMR mapping technique was not available. It is reported in the study limitations and specified in 4.3. paragraph.

4. I think the idea behind parvovirus-19 relation to infarct-like presentation is hypothetically explained by its predilection to endothelial cells, which may lead to endothelial dysfunction and vasospasm causing anginal symptoms with typical infarct-like ECG changes (Marholdt H et la. Circulation 2006) Added in discussion.

5. What are the potential clinical consequences of your results? Would it stop physicians from performing unnecessary angiograms? Probably not. Were there any differences in clinical follow-up? Duration of hospital stay, duration of symptoms, cardiac events? Please include in your analysis as the study is retrospective. This would add clinical value to your study.  In the study limitations is reported that no CMR follow-up examinations were considered to investigate any prognostic implications. It was not the aim of the study which was focused on CMR findings. Other studies (such as DOI: 10.1016/j.ijcard.2016.03.004) investigated the prognostic value of the infarct- and non-infarct like patterns. Duration of hospital stay and duration of symptoms are reported in the new table 1 showing baseline features of the patients.

6. Finally - English needs improvement. For example Line 68 - we instead of were...etc. Corrected.

Round 2

Reviewer 1 Report

The authors have diligently incorporated additional information and made all necessary revisions to enhance the article; however, a minor mistake still remains on page 12, line 339, where the typo 'EF' reappears as 'FE'.

Reviewer 2 Report

The authors have addressed my concerns. I have no further comments. 

None